# A Comparison of Data Reduction Methods for Average Friction Factor Calculation of Adiabatic Gas Flows in Microchannels

**DOI:** 10.3390/mi10030171

**Published:** 2019-02-28

**Authors:** Danish Rehman, Gian Luca Morini, Chungpyo Hong

**Affiliations:** 1Microfluidics Laboratory, Department of Industrial Engineering (DIN), University of Bologna, Via del Lazzaretto 15/5, 40131 Bologna BO, Italy; gianluca.morini3@unibo.it; 2Microscale Heat Transfer Laboratory, Department of Mechanical Engineering, Kagoshima University, Kagoshima Prefecture 890-8580, Japan; hong@mech.kagoshima-u.ac.jp

**Keywords:** underexpansion, Fanno flow, flow choking, compressibility

## Abstract

In this paper, a combined numerical and experimental approach for the estimation of the average friction factor along adiabatic microchannels with compressible gas flows is presented. Pressure-drop experiments are performed for a rectangular microchannel with a hydraulic diameter of 295 μm by varying Reynolds number up to 17,000. In parallel, the calculation of friction factor has been repeated numerically and results are compared with the experimental work. The validated numerical model was also used to gain an insight of flow physics by varying the aspect ratio and hydraulic diameter of rectangular microchannels with respect to the channel tested experimentally. This was done with an aim of verifying the role of minor loss coefficients for the estimation of the average friction factor. To have laminar, transitional, and turbulent regimes captured, numerical analysis has been performed by varying Reynolds number from 200 to 20,000. Comparison of numerically and experimentally calculated gas flow characteristics has shown that adiabatic wall treatment (Fanno flow) results in better agreement of average friction factor values with conventional theory than the isothermal treatment of gas along the microchannel. The use of a constant value for minor loss coefficients available in the literature is not recommended for microflows as they change from one assembly to the other and their accurate estimation for compressible flows requires a coupling of numerical analysis with experimental data reduction. Results presented in this work demonstrate how an adiabatic wall treatment along the length of the channel coupled with the assumption of an isentropic flow from manifold to microchannel inlet results in a self-sustained experimental data reduction method for the accurate estimation of friction factor values even in presence of significant compressibility effects. Results also demonstrate that both the assumption of perfect expansion and consequently wrong estimation of average temperature between inlet and outlet of a microchannel can be responsible for an apparent increase in experimental average friction factor in choked flow regime.

## 1. Introduction

With the pioneering work of Tuckerman and Pease [1], flows in microchannels (MCs) have become of prime importance for heat-exchanging applications. Such systems offer high surface-to-volume ratio and record high heat transfer coefficients with low to moderate pressure drop compared to their macro counterparts [2]. Frictional characteristics of the MC flow are therefore important as they directly translate into pressure drop and hence associated fiscal penalty due to increased required power. Many of earlier experimental pressure-drop studies reported a deviation in the friction factor of MCs compared to macroflows [3,4,5,6,7] while only a few advocated for an agreement [8,9]. A detailed discussion of possible reasons of disagreement with conventional theory has been presented by Morini et al. [10]. The results obtained by testing both liquid and gas flows have confirmed that there exists a good agreement between the correlations used for the prediction of the pressure drop in conventional channels and the experimental data obtained for MCs if no significant scaling effects are present [11]. Due to compressibility effects, gas microflows tend to deviate slightly from macroflow laws which are developed based upon incompressible flows. Numerical results by Hong et al. [12] resulted in a correlation for Poiseuille Number with Mach number which catered for an increased pressure drop due to compressibility compared to the classical law. Experimental results of Yang [13] also showed that the laminar friction factors for microtubes with gas flow are slightly higher than Poiseuille law and follow closely the aforementioned correlation if the Mach number at exit of the microtube is employed for data reduction than its average value along the length. Moreover, most of the earlier experimental studies assumed isothermal flow [14,15,16,17] which holds true for incompressible fluids but not always for compressible gases. Literature has been divided into two main approaches for establishing experimental average frictional characteristics in MCs. When a total pressure drop of MC assembly and inlet temperature are measured, a classical methodology is to invoke minor loss coefficients and subtract minor pressure losses associated with the inlet/outlet manifold from the total measured pressure drop. The resulting pressure difference is then used along with measured temperature at manifold inlet to calculate the average isothermal friction factor. Such a treatment is quite realistic when an incompressible liquid working fluid is tested but the use of this method with compressible flows is questionable. In reality, a gas microflow does not stay isothermal and shows a strong temperature decrease close to MC outlet even for adiabatic walls. In a high-speed gas microflow, placing a thermocouple in the outlet jet will measure a value between static and total temperature [18], and therefore direct measurement is still challenging. Fortunately for an adiabatic flow, static temperature estimation at MC outlet can be done using a quadratic equation proposed by Kawashima and Asako [18]. By incorporating temperature gradient along the length of MC, it was shown numerically that local friction factor for nitrogen gas in microtubes (MTs) is in good agreement with conventional theory not only in laminar but also in the high-speed turbulent flow regime. Hong et al. [19] later measured local pressure at five axial positions of rectangular MCs and considered gas temperature change for the experimental calculation of semi-local friction factors (i.e., between two closely placed pressure ports). Semi-local turbulent friction factors considering Fanno flow were in good agreement with Blasius law whereas they were lower when gas was considered isothermal. In this study; however, authors did not calculate average friction factor between inlet and outlet of microchannel as the case with most of the previous experimental studies. Recently a new data reduction methodology for the average friction factor calculation between inlet and outlet has been developed by Hong et al. [20]. Experimental results for 867 μm MT vented to atmosphere showed better agreement with the Blasius law in the turbulent flow regime using an improved equation for adiabatic friction factor. Applicability of this new data reduction methodology on a rectangular channel with higher degree of flow choking is discussed in the present work.

From the recent literature, it can be established that most of the earlier studies used minor loss coefficients to calculate experimental average friction factors between inlet and outlet of a MC/MT. Furthermore, almost all such studies assumed an isothermal flow inside MC. On the contrary, a few studies that dealt with temperature change for compressible gas flows avoid the use of minor loss coefficients and present the evaluation of semi-local friction factors instead. With MC/MT, it is not trivial to obtain local pressure and temperature values, but it has been demonstrated how in the presence of significant compressibility effects, an accurate estimation of the friction factor is feasible only by assuming non-isothermal gas flow along the channel. If the local temperature trend cannot be measured experimentally, data reduction method for the accurate estimation of the adiabatic friction factor for compressible gases must be based on a series of assumptions that are able to predict this variation along the MC. Current study aims to demonstrate which assumptions must be adopted for a better evaluation of experimental friction factor by showing the difference in terms of average friction factors that are obtained by adopting different data reduction methods proposed in literature. Moreover, by coupling numerical modeling with experimental results, the effect of gas flow choking on evaluation of experimental average friction factor is also elucidated. Finally, a unique methodology for the analysis of pressure drop in presence of compressible gases in MCs is suggested to the reader.

## 2. Experimental Methodology

### 2.1. Setup

Schematic of test bench and MC assembly used in this work are shown in Figure 1. Nitrogen gas is stored in a high-pressure flask (①, 200 bar) and brought to approximately 11 bars and ambient temperature before entering into a 7 μm particle filter (②, Hamlet^®^), used to prevent possible impurities from clogging the MCs or the flow controller. The facility is equipped with three volume flow rate controllers (Bronkhorst EL-Flow E7000) operating in the 0–50 NmL/min (4a), 0–500 NmL/min (4b) and 0–5000 NmL/min (4c) ranges respectively. A three-way valve (③) then directs the working fluid to the proper flow controller by means of a computer-steered valve. This allows to impose a certain volume flow rate through the MC to achieve desired Reynolds number. Gas is then allowed to enter the MC test assembly (⑤). The total pressure drop between the inlet and the outlet of the MC assembly is measured by means of a differential pressure transducer (⑥, Validyne DP15) with an interchangeable sensing element that allows accurate measurements over the whole range of encountered pressures. Atmospheric pressure is measured using an absolute pressure sensor (⑦, Validyne AP42). To measure the temperature at the entrance of the MC a K-type, calibrated thermocouple is used (⑧). Thermocouple voltage and an amplified voltage of pressure sensors are fed to internal multiplexer board of Agilent 39470A and are read by means of a PC using a Labview^®^ program.

The MC is fabricated by milling a PMMA plastic sheet with a nominal thickness of 5 mm. CNC milling is performed using Roland^®^ MDX40A with a 100 W spindle motor. Modeling and CNC toolpath are generated in Autodesk^®^ Fusion360. A flat end mill of 300 μm is used to make MC slot. Dry milling is performed using spindle rpm and speed of 10,000 rpm and 300 mm/min, respectively. A constant depth of cut of 100 μm is used. Dimensions and inner surface roughness of resulting channel are measured using an optical profilometer. Cross section and roughness measurements are performed at three locations along the length of the MC and an average of 3 readings is taken at each cross section. The average width (*w*), height (*h*) and surface roughness (ϵ) of realized MC are 360 μm, 250 μm and 1.05 μm, respectively. PMMA chip containing MC and a microscopic view of bottom and top of the MC is shown in Figure 2. Lower PMMA sheet containing milled MC is sealed by means of a top cover of same material and an O-ring as shown in Figure 1b. Both top and bottom plastics are sandwiched between two 5 mm thick Aluminum plates. Whole assembly is then bolted to ensure leak tightness. Test section is checked for leakage by applying a pressure of almost 10 bars between the inlet and outlet connectors and closing the outlet manifold. Pressure inside the assembly is monitored for at least 5 h to spot any major leakage. Such a test is repeated before initiating experimental test campaign. Small holes of 200–250 μm are drilled in top plastic at 3 axial locations x=0.58L,0.72L and 0.87L. Local static pressure is read through these holes using a solenoid valve-switching assembly. At each pressure tap, differential pressure between the port and atmosphere is measured using the differential pressure gauge. The atmospheric pressure is also measured at every data point and is finally added to differential reading to obtain absolute pressure of the specific cross section. Typical uncertainties associated with instruments used in current experimental work are reported in Table 1.

### 2.2. Data Reduction

Local Fanning friction factor can be defined by the following expression for a compressible flow [18]:(1)$$f_{f,local}=\frac{4\tau_{w}}{\frac{1}{2}\rho u^{2}}=\frac{2D_{h}}{p}-\frac{2D_{h}p}{\rho^{2}u^{2}RT}\frac{dp}{dx}-\frac{2D_{h}}{T}\frac{dT}{dx}$$
where hydraulic diameter of a rectangular MC is defined as:(2)$$D_{h}=\frac{4A}{Per}=\frac{2wh}{w+h}$$
Reynolds number at the inlet of MC can then be calculated using measured mass flow rate and calculated viscosity at inlet temperature with the following equation:(3)$$Re=\frac{\dot{m}D_{h}}{\mu A}$$
Considering one dimensional flow of ideal gas, Equation (Equation 1) can be integrated between two points a and b along the length (*L*), to calculate average friction factor between those points as follows:(4)$$f_{f}=\frac{D_{h}}{x_{b}-x_{a}}\left[\frac{p_{a}^{2}-p_{b}^{2}}{RT_{av}{\dot{G}}^{2}}-2\ln\left(\frac{p_{a}}{p_{b}}\right)+2\ln\left(\frac{T_{a}}{T_{b}}\right)\right]$$
For the rest of the text, when Equation (Equation 4) is applied between two closely spaced pressure ports (e.g., between $p_{4}$–$p_{5}$ in Figure 3 for $\Delta x=x_{5}-x_{4}$), resulting friction factor will be referred to as semi-local friction factor ($\tilde{f_{f}}$) while when it is applied between inlet and outlet of MC (Δx=L), it will be known as average adiabatic friction factor ($f_{f}$). In addition, under the hypothesis of adiabatic compressible flow, the energy balance for one dimensional Fanno flow, between inlet ’*in*’ and any other cross section at a distance ’*x*’ from the inlet of the MC yields the following quadratic equation for the estimation of average cross-sectional temperature [18]:(5)$$\left(\frac{\rho_{in}^{2}u_{in}^{2}R^{2}}{2c_{p}p_{x}^{2}}\right)T_{x}^{2}+T_{x}-\left(T_{in}+\frac{u_{in}^{2}}{2c_{p}}\right)=0$$

Finally, knowing the average pressure and temperature of a specific cross section, average density and velocity of compressible gas can be obtained using gas and continuity equations, respectively. The local value assumed by Mach number, defined as the ratio of velocity and the local speed of sound, can be calculated as follows:(6)$$Ma_{x}=\frac{u_{x}}{\sqrt{\gamma RT_{x}}}$$
In all the published experimental results, MC/MT is attached to a conventional piping system using an entrance manifold. The geometry of the manifold may vary case by case but a pressure drop between the manifold and MC inlet exists there. Similarly, when gas exits the MC/MT, the expansion of the gas to the exit manifold or atmosphere causes an additional pressure drop. This is shown schematically in Figure 3. The current experimental MC assembly has a reducer that connects the entrance manifold to the gas supply piping. Similarly, another reducer towards the exit of assembly vents the gas to the atmosphere coming from exit manifold. Therefore, minor losses ($\Delta P_{in/out}$) in reducers and manifolds need to be accounted for to estimate the pressure drop of the MC/MT, alone ($\Delta P_{ch}$ see Figure 3) from the total measured pressure drop.

The most used method for estimating these minor losses is to use loss coefficients ($K_{in/out}$) available in fluid mechanics texts, which are generally validated for liquids and weakly compressible gases. Minor pressure losses are defined as [21]:(7)$$\Delta P_{in/out}=K_{in/out}\frac{1}{2}\rho u_{in/out}^{2}$$
Data reduction methodology where numerical inlet minor loss coefficients are used along with the temperature estimation at MC outlet using Equation (Equation 5), is referred to as M1 in the subsequent text. An alternative methodology (M2), followed by the group of Prof Asako is to estimate MC inlet flow properties by assuming isentropic flow between the manifold and MC inlet. This automatically caters for a reduction in MC inlet pressure and hence the use of $K_{in}$ is no more required. An initial estimate of the gas velocity at MC inlet is made as in M1 (i.e., using the measured mass flow rate and density of the gas at the inlet of assembly). Inlet properties are then calculated iteratively using the following set of equations (see Figure 3):(8)$$T_{in}=T_{1}-\frac{{u_{in}}^{2}}{2C_{p}}$$
(9)$$p_{in}=\frac{p_{1}}{{\left(1+\frac{{u_{in}}^{2}}{2C_{p}T_{in}}\right)}^{\frac{\gamma }{\gamma -1}}}$$
(10)$$\rho_{in}=\frac{p_{in}}{RT_{in}}$$
(11)$$u_{in}=\frac{\dot{m}}{\rho_{in}A}$$

Main differences between the two experimental data reduction methods (M1 and M2) described before are summarized in Table 2. In the next sections, experimental results for average friction factor are deduced by using both M1 and M2 approaches. These results are then compared with numerical predictions to put in evidence discrepancies among experiment and theory. This comparison is finally used to establish the most accurate data reduction procedure for the estimation of average friction factor for MC/MT in presence of compressible gases.

## 3. Numerical Methodology

Due to small dimensions of MC assemblies, it is not possible to insert as many sensors as one desires along a MC. To overcome this lack, a validated CFD model can be used to gain an insight of flow physics [11]. Therefore, a numerical model of the experimental test assembly is developed in ANSYS Workbench framework. Three MC dimensions are simulated in the current study as tabulated in Table 3. MC1 is used to replicate the channel tested experimentally and gain insight of flow physics by comparing with experiments whereas MC2 is used for the validation of adapted numerical scheme. Simulation results of aforementioned MCs along with MC3 are also used for discussing the role of loss coefficients in the calculation of friction factor. An exhaustive parametric study to individuate effects of $D_{h}$ and aspect ratio ($\alpha =\frac{h}{w}$) on minor losses evaluation, is out of scope for current analysis. Therefore, a limited set of simulations with only 3 MCs with different hydraulic diameters and aspect ratios are chosen to emphasize minor losses dependence on the MC dimensions.

Geometry and meshing is done using Design Modeler and ANSYS Meshing software, respectively. A mesh of 45×30×200 is used in the MCs. A structured mesh locally refined at the walls of the channel and manifolds is employed as shown in Figure 4. The mesh expansion factor is kept as 1.1 and first node point is placed such that y+, which is non-dimensional distance between first mesh node and MC wall, is in between 1 and 4 for the highest Re simulated. Orthogonality of mesh elements inside MC is between 0.95–1 in all the simulated cases. A commercial solver CFX based on finite volume methods is used for the flow simulations. Reducers and manifolds are also simulated along with the MCs. Height of the reducer ($H_{r}$, see Figure 4) is 30 mm with an internal diameter of 4 mm. Whereas diameter of the circular manifolds is 9 mm and height ($H_{man}$) is kept same as the height of the simulated MC ($H_{man}=h$). Dimensions of these parts are chosen based upon the experimentally tested MC assembly. Ideal nitrogen gas enters the entrance manifold that is orthogonal to MC and leaves again orthogonally through the exit manifold. For the MC that is also experimentally tested (MC1), measured mass flow rate is used while for other cases it is calculated from Equation (Equation 3).

Steady state RANS simulations are performed for all turbulent cases. Laminar flow solver is used for the cases where Re≤1000 and for Re>1000, SST *k*-ω transitional turbulence model is used. A modified formulation of γ-$Re_{\theta }$ transition turbulence model for internal flows is applied [22]. High-resolution turbulence numerics are employed with a higher order advection scheme available in CFX. Pseudo time marching is done using a physical timestep of 0.1s. A convergence criteria of $10^{-6}$ for RMS residuals of governing equations is chosen while monitor points for pressure and velocity at the MC inlet and outlet are also observed during successive iterations. In case where residuals stayed higher than supplied criteria, the solution is deemed converged if monitor points did not show any variation for 200 consecutive iterations. Reference pressure of 101 kPa was used for the simulation and all the other pressures are defined with respect to this reference pressure. Energy equation was activated using total energy option available in CFX which adopts energy equation without any simplifications in governing equations solution. Kinematic viscosity dependence on gas temperature is defined using Sutherland’s law. Further details of boundary conditions can be seen in Table 4.

To estimate friction factor and minor loss coefficients, five different cross-sectional planes are defined at x/L of 0.005, 0.25, 0.5, 0.75 and 0.98, respectively. In addition, two planes defined at x/L of 0.0005 and 0.9995 are treated as the inlet and outlet of MC, respectively. Results from these planes are further post processed in MATLAB to deduce required flow quantities. Numerical friction factors are then evaluated simply by using Equation (Equation 4).

### Validation

To validate the adopted numerical scheme, a MC with hydraulic diameter of 203 μm (MC2) is simulated and numerical friction factors are compared with experiments of Hong et al. [19]. The width and height of the MC are 1020 μm and 112.7 μm respectively giving it an aspect ratio of 0.11. Length of the channel in numerical model is taken as 100 mm while it is 26.9 mm in experiments performed by [19]. Moreover, dimensions of the inlet manifold are slightly different in experimental settings than those adapted in numerical model. Since a comparison of semi-local friction factor ($\tilde{f_{f}}$) towards the last half of the MC is made, these differences should not induce a significant effect on $\tilde{f_{f}}$ in that region.

Hong et al. [19] reports semi-local friction factor measured between two closely placed pressure ports at the dimensionless length (x/L) of 0.67–0.8. Obtained numerical results from x/L= 0.5–0.75 (o) and x/L= 0–1 (Δ) are compared to the experimentally reported values in Figure 5. There exists an excellent agreement between the current numerical results and experimental results in the laminar flow regime where $f_{f}$ follows the Shah & London correlation (S&L):(12)$$f_{f_{SL}}=\frac{96}{Re}\left(1-1.3553\alpha +1.9467\alpha^{2}-1.7012\alpha^{3}+0.9564\alpha^{4}-0.2537\alpha^{5}\right)$$
In the turbulent flow regime, both experimental and numerical results are slightly above the Blasius law. It is to be noted that even by assuming smooth walls in the numerical simulation, the turbulent friction factor can be slightly higher than Blasius law which is also the case of experimental results of semi-local friction factor reported by [19]. Therefore turbulent $f_{f}$ can be higher than Blasius law even with smooth walls, if compressible effects are significant.

## 4. Results and Discussion

### 4.1. Numerical Calculation of Minor Loss Coefficients

The minor loss associated with the inlet from numerical results is calculated using Equation (Equation 7) between the inlet of assembly and MC inlet and therefore includes 90${}^{\circ }$ bend loss due to orthogonal flow direction change at the manifold inlet. Results of $K_{in}$ and $K_{out}$ for the three MCs are compared in Figure 6. $K_{in}$ decreases steeply in laminar and early turbulent flow regimes, whereas it becomes almost a constant in high turbulent flow regime as shown in Figure 6a. At the lower Re, MC assembly pressure drop in experimental campaign is also usually lower and hence assuming a smaller and/or constant $K_{in}$ would certainly cause $f_{f}$ to be higher than macro theory. For the smallest α simulated, $K_{in}$ is as high as 5.19 which is significantly higher than values available in general fluid mechanics text [21]. For a rectangular MC, $K_{in}$ is a function of α and $D_{h}$ simultaneously and hence it must be evaluated numerically in advance to help in experimental data reduction. $K_{out}$ on the other hand stays close to zero in laminar and early turbulent flow regime and shoots rapidly in higher turbulent flow regimes as seen in Figure 6b. From the investigated MC assembly, it is evident that $K_{out}$ is highest for the smallest $D_{h}$ simulated for high turbulent flow regime and decreases with an increase of $D_{h}$. Numerical results show that $K_{out}$ for compressible flows, can also go higher than its limiting value of 1 as calculated using the following theoretical relation:(13)$$K_{out,th}={\left(1-\frac{A_{MC}}{A_{man}}\right)}^{2}$$
where $A_{MC}$ and $A_{man}$ denote cross-sectional area of MC and manifold, respectively.

A MC assembly similar to that considered in the present work has been tested by Mirmanto [23]. The author used a conventional value of 1.2 as loss coefficient for 90${}^{\circ }$ bend which is not observed in the present numerical results. Similar assembly has also been investigated numerically by Sahar et al. [24] where inlet losses in laminar incompressible flow increased for increasing $D_{h}$ for a constant value of aspect ratio α. A need for systematic investigation of minor losses using numerical modeling as a priori is also emphasized in [25]. The three MCs modeled in this paper demonstrated that minor loss coefficients are dependent on assembly as well as MC geometry and in all cases are not equal to general values used in literature for macro flows and hence a numerical model is required for an accurate estimation friction factors especially in laminar flow regime. This is due to the fact that minor losses carry a significant portion of the total pressure drop in laminar flow regime but as the Re is increased, the contribution of minor losses diminishes, and they do not have significant effect on calculation of $f_{f}$ in turbulent flow regime.

### 4.2. Experimental Average Friction Factor

Pressure-drop experiments are performed for MC1 using nitrogen gas as working fluid. Results in terms of $f_{f}$ in laminar flow regime are compared with Shah & London correlation (Equation (Equation 12)) for rectangular channels while Blasius law in turbulent flow regime is used for comparison. Laminar $f_{f}$ is also compared with the correlation of Hong et al. [12] that caters for an increase in laminar $f_{f}$ due to compressibility. Numerically obtained values of $K_{in}$ are used in data reduction M1. When outlet is open to atmosphere, a convenient practice is to estimate the experimental $f_{f}$ by assuming a fully expanded flow at the MC outlet. This assumption essentially makes $K_{out}$ to be zero and is quite realistic in the laminar and early turbulent flow regimes as shown numerically in Figure 6b. Therefore, due to the lack of MC outlet pressure measurement in the current experimental campaign, $K_{out}=0$ is used in further data reduction. Limitations to this assumption will be discussed later in the text.

Previous section already put in evidence that minor losses differ for different MC geometries. Therefore, for an accurate evaluation of inlet minor losses a curve fit on the numerical results is used to extract the variation of $K_{in}$ with Re. This is used to model the inlet minor loss to calculate $f_{f}$ with data reduction M1. The adiabatic $f_{f}$ calculated using Equation (Equation 4) for both methodologies M1 and M2 is shown in Figure 7. There is an excellent agreement between experimental and numerical $f_{f}$ in laminar flow regime where both follow the Hong et al. [12] correlation within experimental uncertainty. Uncertainty bars are omitted in Figure 7 for reasons of clarity. Isothermal friction factor ($f_{f,iso}$) obtained by assuming $T_{in}=T_{out}$ and hence neglecting the last term of Equation (Equation 4), stays lower than Blasius law in turbulent regime. Results of numerical $f_{f}$ are slightly higher than Blasius in high turbulent regime. Experimental $f_{f}$ in early turbulent flow regime is lower than Blasius law with both M1 and M2 methods and starts not only increasing again in high turbulent flow regime but with a slope that diverges from Blasius at Re > 10,000. On the contrary, slope for numerical $f_{f}$ does not diverge significantly from Blasius law in turbulent flow regime. Since the relative roughness ($\frac{\epsilon }{D_{h}}$) of the tested channel is equal to 0.5%, such an increase in experimental $f_{f}$ compared to Blasius law using both M1 and M2 methods can potentially be associated with the rough channel walls. To better investigate this aspect, experimentally deduced flow properties are further compared with numerical ones.

The temperature at the inlet and outlet of MC is compared in Figure 8a. Discrepancy in the inlet temperature for M1 method is a direct consequence of the assumption of constant density and temperature at MC inlet, both assumed equal to the measured values at the manifold inlet. In absolute sense, there is a maximum difference of 7K in turbulent flow regime. Gas stays almost isothermal at low Reynolds numbers (Re < 1000) but as Re is increased, relative pressure difference between the manifold inlet and MC inlet increases causing a decrease in temperature at MC inlet that is not captured in M1 method. Temperature at the outlet estimated using Equation (Equation 5) follows the numerical trend where the gas temperature is decreasing with increasing Re. For such an estimation, an absolute temperature difference of approximately 26K compared to 92K in case of isothermal assumption at Re of around 13,000 is observed. This difference furthers as Re is increased. On the other hand, when the MC inlet conditions are defined using M2 method, inlet expansion and hence temperature decrease from the entrance manifold to MC inlet is correctly modeled as shown in Figure 8a. This result suggests using M2 method for correctly predicting the MC inlet flow properties without using an additional $K_{in}$. Outlet flow properties, however, differ from the numerical values for both M1 and M2 methods. It is worth mentioning that outlet pressure is assumed to be atmospheric as outlined in Table 2 ($K_{out}=0$) for both methods to estimate outlet temperature. Numerical value assumed by the Mach number at the MC outlet is shown in Figure 8b where it keeps on increasing with Re and finally gets to an almost constant value after Re =10,000. At this point, flow starts to choke and Ma at the outlet reaches a constant value of close to 1 (i.e., in this case $M_{a}=1.19$ after Re = 15,000). An explanation of supersonic jet at the exit of constant area ducts has been presented by Lijo et al. [26]. Numerical works of Kawashima et al. [27] and Hong et al. [28] showed that Ma at the outlet can go higher than its maximum limit of 1(Fanno flow). This happens due to shear thinning of the boundary layer close to the outlet of MC/MT that serves as de-Laval nozzle for incoming high subsonic jet of gas flow. Flow choking at the outlet is not captured by experimental data reduction methodologies M1 and M2 and therefore Ma keeps on increasing with Re in both methods. This result disagrees with what is seen in numerical results. At maximum Re values, outlet Ma reaches as high as 2.46 for isothermal treatment of gas while it reduces to 1.88 in cases of M1 and M2 methods. These very high values of Ma signify that both methods lack the flow physics at the outlet. This is due to the wrong estimation of outlet jet temperature using Equation (Equation 5) at very high Re as can be seen in Figure 8a. Experimental data shows flow choking at the inlet where Ma becomes constant for Re>6000. A constant temperature and Ma ensure a constant velocity at MC inlet which explains the reason for a constant value of inlet loss coefficient in the turbulent flow regime after an initial decrease in laminar and early the turbulent flow regime, as observed in Figure 6a. However, it is evident that flow choking at inlet starts much earlier than outlet and numerical results of MC inlet temperature are lower than assumed in M1. Effect of flow choking on calculation of average $f_{f}$ is discussed in the next section.

### 4.3. Flow Choking and under Expansion at Outlet

Usually in fluid mechanics text, flow choking is defined as the flow condition in which working fluid at outlet of a pipe attains a sonic velocity and therefore mass flow rate cannot be increased any further. However, as shown by the current experiments and by the work of Hong et al. [28], mass flow rate for a compressible working fluid keeps on increasing even when the flow is choked. Ma and hence velocity, however, attains a maximum at all cross sections of the pipe in choked gas flows. This is simply because of an increase in upstream density of the gas by increasing upstream pressure causes mass flow rate to increase even if velocity is choked downstream. Definition of flow choking shall be given therefore by highlighting the presence of maximum in Ma (velocity) rather than in mass flow rate for compressible flows. Variation of numerical outlet Ma with ratio of MC outlet pressure and atmospheric pressure is shown in Figure 9a for all three simulated MCs. It can be seen that as soon as outlet Ma starts to choke, MC outlet pressure starts increasing higher than atmospheric pressure which means that the flow at MC outlet becomes underexpanded. The measure of underexpansion is not much pronounced during the experimental tests on MC1 where flow choking begins around Re=9934. On the contrary, flow choking is observed earlier in MC2 and MC3 in correspondence of Re=5000 and Re=3999 respectively. After these Re, outlet pressure shoots above the atmospheric pressure and the flow cannot be considered as fully expanded as assumed in both M1 and M2 methods. As a result, a sudden increase in outlet numerical $K_{out}$ after flow starts to choke can be noted in Figure 9b. As discussed earlier, a single value of the outlet loss coefficient is therefore not enough, especially for the MCs with smaller hydraulic diameter where flow choking can start even in laminar flow regime.

From experimental point of view, it is difficult to measure static pressure exactly at the MC outlet and therefore an assumption of fully expanded flow serves the purpose. However, as evidenced in all three simulated cases, this holds true for unchoked gas flows only. Moreover, the measure of underexpansion becomes stronger with smaller dimensions of the channel and hence should be of concern for deducing friction factors using Equation (Equation 4) in any experimental campaign. As Equation (Equation 5) requires cross-sectional average pressure to estimate the average temperature of that cross section, full expanded flow assumption therefore always overestimates the decrease in outlet temperature in choked gas flows. This in turn causes an artificial increase in the calculation of friction factor in the choked flow regime from experimental pressure-drop data. Therefore experimental $f_{f}$ shows a deviation from Blasius law in Figure 7 while numerical results agree with Blasius.

Flow choking that was observed in numerical data can experimentally be established only using local flow properties. For an adiabatic flow in a duct, fluid temperature and density at a specific cross section can be estimated using only a static pressure measurement. Thus, local pressure measurement will result in a better understanding of choked compressible flow along the length of MC. A comparison of local measured static pressure with numerical results is shown in Figure 10a for MC1 and a zoomed portion of laminar to early turbulent flow regime is shown in Figure 10b. Numerically estimated values are generally in good agreement with experimental local pressure values in laminar and early turbulent regimes. In highly turbulent flow regime (Re > 10,000) prescribed boundary conditions result in an overall lower pressure drop than what is experimentally observed. However, such difference is not too significant and therefore numerical model can be used as a priori tool to have first estimation of pressure drop and local flow physics (especially flow choking).

Numerical local temperature is also in good agreement with the estimated cross-sectional temperature obtained by using Equation (Equation 5) as can be seen in Figure 11a. Experimental flow choking of the gas flow is also evident in Figure 11b where numerical and experimentally deduced Ma are in excellent agreement. It is, therefore, not erroneous to assume that numerical flow choking at outlet where Ma goes higher than 1 (as in the case of MC1), would also be encountered in experiments. It is interesting to note that all through the data reduction for the results presented in this section, no numerical input has been considered as a priori and MC inlet properties are calculated by considering an isentropic expansion between the entrance manifold and MC inlet.

Another interesting observation from Figure 7 is that in choked flow regime, isothermal treatment of gas results in $f_{f,iso}$ that although is lower than Blasius law but follows its slope. The same is true for semi-local $\tilde{f_{f}}$ between last two pressure taps at x/L= 0.72–0.87 which again follows the slope but stays slightly higher than Blasius law. On the contrary $f_{f}$, using both M1 and M2 methods shows a false increase when the flow chokes. This again points towards a possibility of a wrong estimation of average gas temperature assumed in M1 and M2 methods. Because such a rapid increase in slope of $f_{f}$ is not observed when difference in gas temperature is relatively small as is the case for semi-local values (see Figure 13a) or when it is zero (for isothermal gas treatment). In fact, if the numerical static temperature is analyzed along the length of MC as shown in Figure 12a, it can be established that even for the highest simulated Re, temperature almost stays isothermal along most of the length of the MC with a sudden decrease towards the end of MC. Therefore, to calculate average $f_{f}$ between inlet and outlet, an equal weighted average of measured inlet temperature and estimated at the outlet using Equation (Equation 5) would underpredict the real average gas temperature. Average temperature of the gas between two pressure ports ’*a*’ and ’*b*’ along the length of MC can be defined by:(14)$$T_{av}=c_{1}T_{a}+c_{2}T_{b}$$
The effect of values of $c_{1}$ and $c_{2}$ on temperature average between inlet and outlet of MC and hence evaluation of $f_{f}$ is shown in Figure 12b. It is evident that friction factor is more in agreement with Blasius law if the average is obtained with $c_{1}>c_{2}$.

Therefore, in absence of measured outlet pressure it is appropriate to estimate the average temperature by assuming a higher weight of the inlet temperature with respect to the outlet temperature (Equation (Equation 5)), obtained by considering the fully expanded flow assumption. For a case where pressure taps are located close to each other such that there is no significant temperature change from one tap to the other, average temperature in between can be approximated well with an equal weighted average ($c_{1}=c_{2}=0.5$) and therefore semi-local $\tilde{f_{f}}$ follows Blasius law even in the choked turbulent flow regime. However, if the distance between these ports is large (this is the case with inlet and outlet of MC) an integral average of temperature between these ports should be considered. A detailed derivation of the average friction factor equation using such temperature average is given in [20] and therefore it is skipped in this text. The calculation of average friction factor for Fanno flow (adiabatic walls) between two pressure ports *a* and *b* can be obtained as follows:(15)$$\begin{array}{c} f_{f,av}=\frac{D_{h}}{x_{b}-x_{a}}[\left(-2\ln\frac{p_{a}}{p_{b}}+2\ln\frac{T_{a}}{T_{b}}\right)-\left(\frac{1}{{\dot{G}}^{2}R\left(T_{in}+\frac{{u_{in}}^{2}}{2c_{p}}\right)}\right)\times \\ \left(\frac{{p_{b}}^{2}-{p_{a}}^{2}}{2}+\frac{B^{2}}{2}\ln\frac{p_{b}+\sqrt{p_{b}^{2}+B^{2}}}{p_{a}+\sqrt{p_{a}^{2}+B^{2}}}+\frac{1}{2}\left(p_{b}\sqrt{p_{b}^{2}+B^{2}}-p_{a}\sqrt{p_{a}^{2}+B^{2}}\right)\right)] \end{array}$$
where $B^{2}=4\beta \times \frac{{\dot{G}}^{2}R^{2}}{2c_{p}}\times \left(T_{in}+\frac{u_{in}^{2}}{2c_{p}}\right)$ and β, kinetic energy recovery coefficient, is taken as 2 for laminar and 1 for turbulent flow.

Semi-local $\tilde{f_{f}}$ curves between x/L=0.58−0.72 and x/L=0.72−0.87 are shown in Figure 13a for MC1 experiments. Average $f_{f}$ between the inlet and outlet using Equations (Equation 4) and (Equation 15) is also plotted for comparison. There is an excellent agreement between $f_{f,av}$ calculated using Equation (Equation 15) and Blasius law even in choked turbulent flow regime. This is due to the fact that an integral average of the temperature is used instead of an equal weighted average between the inlet and outlet of MC. Whereas average friction factor calculated using Equation (Equation 4) suffers a deviation from Blasius law only due to erroneous estimation of the bulk fluid temperature between the inlet and outlet of MC. It is reminded to the reader that in Figure 13a, MC outlet pressure is still assumed to be the atmospheric as was done with M1 and M2 methods to evaluate $f_{f,av}$. However, as Equation (Equation 15) does not require for an explicit approximation of $T_{av}$ between inlet and outlet of MC, resulting $f_{f,av}$ values are therefore in a better agreement with Blasius law. To emphasize it further, results of average friction factor by Equations (Equation 4) and (Equation 15) with numerical estimated pressure (num. $p_{out}$) instead of atmospheric pressure at outlet, are also shown in Figure 13b. A significant improvement in the slope of turbulent $f_{f}$ calculated using Equation (Equation 4) can be seen when numerical outlet pressure is used due to better estimation of outlet temperature. Results however, are still higher than values calculated using Equation (Equation 15) because an equal weighted $T_{av}$ between inlet and outlet (c1=c2=0.5) is assumed. On the contrary, $f_{f,av}$ with numerical outlet pressure calculated using Equation (Equation 15) is in closer agreement with Blasius law mainly due to better estimation of average temperature of gas. Therefore Equation (Equation 15) is recommended for calculating average experimental $f_{f}$ between two pressure ports for adiabatic MCs. Equation (Equation 4) should be considered as an approximation of Equation (Equation 15) for two closely placed pressure taps, where temperature change between these taps can be well represented with an equal weighted average of respective temperatures, as is the case for semi-local friction factor calculations. An experimental/numerical campaign to analyze the applicability of Equation (Equation 15) on MCs of smaller $D_{h}$ with even higher degree of flow choking is therefore recommended to complete this analysis.

To summarize the results and discussion of current work following recommendations are being made:In laminar flow regime, due to their significant relative contribution towards pressure drop, minor losses can only be estimated using validated numerical results for compressible gas microflows. To avoid numerical input while deducing experimental friction factor results, M2 method can be used to correctly model pressure and temperature at the inlet of MC.In choked turbulent regime, MC outlet pressure should be measured to better estimate the cross-sectional average temperature using Equation (Equation 5). For an experimental campaign where it is not possible, adiabatic friction factor with an assumption of perfect expansion ($p_{out}=p_{atm}$) should be evaluated using Equation (Equation 15).

## 5. Conclusions

From the above-mentioned discussion followings points can be inferred:For the correct experimental estimation of $f_{f}$ from total pressure drop and inlet temperature measurements employing minor losses methodology proposed for incompressible flows (M1), it is essential to have inlet and outlet loss coefficients calculated numerically as a priori else results will be misleading in the presence of strong compressibility effects.Gas flow properties at MC inlet can be obtained by considering an isentropic expansion of gas between the entrance manifold and MC (M2). This, when coupled with the assumption of perfect expansion at MC outlet, results in a self-sustained experimental data reduction method with no numerical input as a priori. Furthermore, this avoids possibility of inducing errors in inlet minor loss estimation due to poor numerical modeling.While deducing experimental results for $f_{f}$ between the inlet and outlet, perfect expansion ($K_{out}=0$) can be used as a first approximation till the flow starts to choke at outlet. After this limit, gas flow is underexpanded and a fully expanded treatment will result in a significant overestimation of MC outlet temperature using Equation (Equation 5) and hence an artificial increase of $f_{f}$ in choked flow regime is observed.Detailed experimental and numerical analysis shows that the gas flow can be assumed isothermal only in the laminar flow regime for the evaluation of friction factors in MCs.Experiments also show that the isothermal treatment of the gas results in friction factors that are usually lower than adiabatic ones in the choked turbulent flow regime with $f_{f,iso}$ following the slope of Blasius law and adiabatic $f_{f}$ diverging from it at higher Re. The reason for this diversion is the inappropriate data reduction at MC outlet during flow choking regime.As gas flow accelerates towards the end of the MC causing a steep decrease in temperature near the outlet, average friction factors between inlet and outlet should be calculated using Equation (Equation 15).

## Figures and Tables

**Figure 1 micromachines-10-00171-f001:**
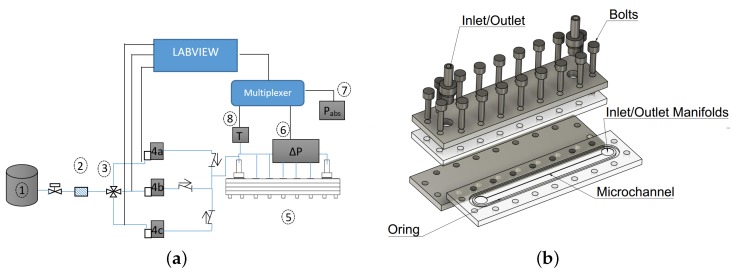
Experimental setup (**a**), and an exploded view of MC assembly (**b**).

**Figure 2 micromachines-10-00171-f002:**
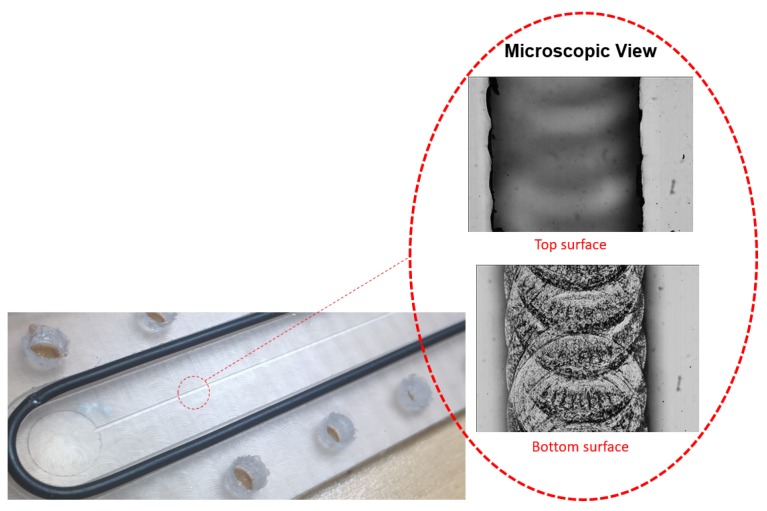
Zoomed part of chip containing MC and its top and bottom surfaces.

**Figure 3 micromachines-10-00171-f003:**
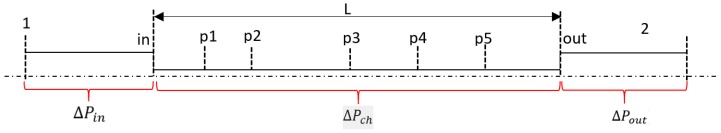
Schematic of minor losses.

**Figure 4 micromachines-10-00171-f004:**
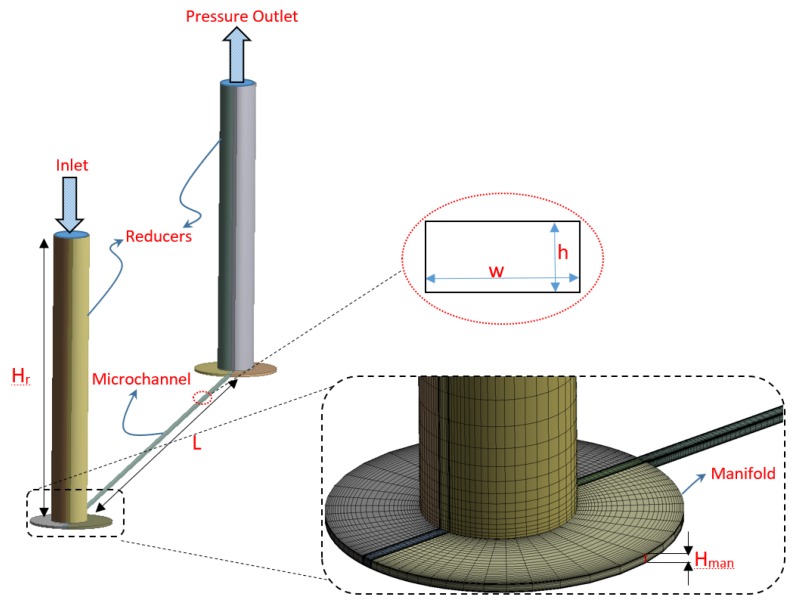
Details of numerical model and meshing.

**Figure 5 micromachines-10-00171-f005:**
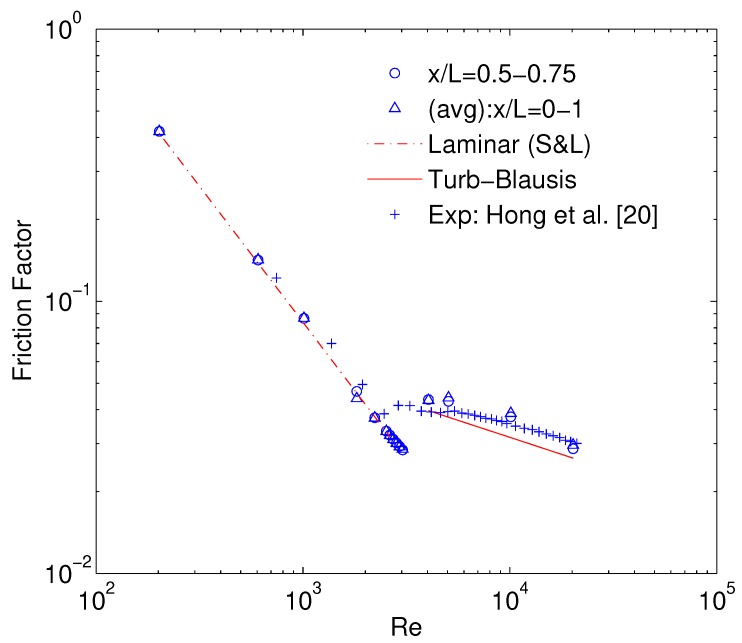
Numerical validation of friction factor calculation.

**Figure 6 micromachines-10-00171-f006:**
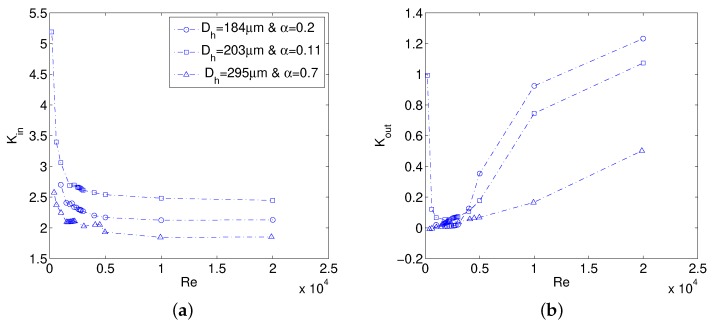
Numerical minor loss coefficients: $K_{in}$ (**a**), and $K_{out}$ (**b**).

**Figure 7 micromachines-10-00171-f007:**
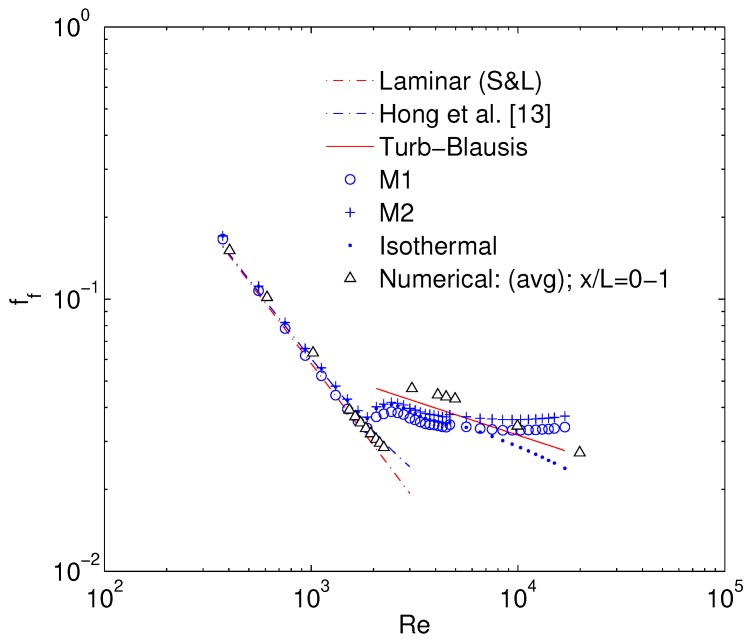
Comparison of numerical and experimental $f_{f}$ using M1 and M2.

**Figure 8 micromachines-10-00171-f008:**
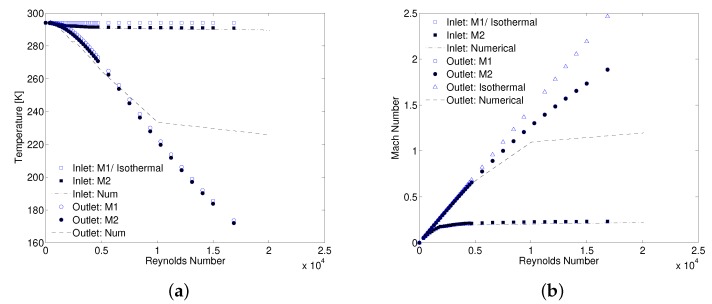
Comparison of flow properties using M1 & M2: local temperature (**a**), and Mach number (**b**).

**Figure 9 micromachines-10-00171-f009:**
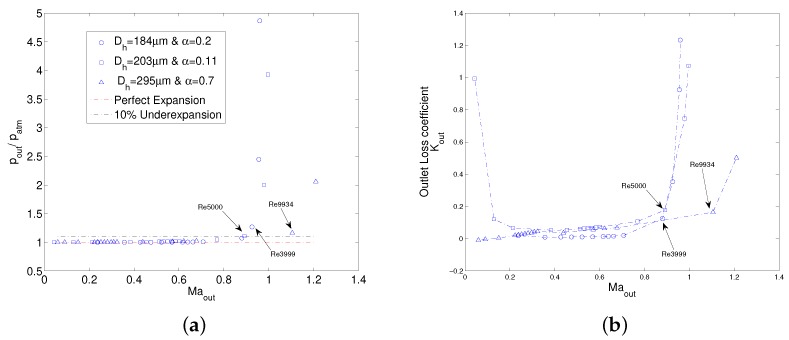
Flow choking and underexpansion for simulated MCs: back pressure (**a**), and $K_{out}$ (**b**).

**Figure 10 micromachines-10-00171-f010:**
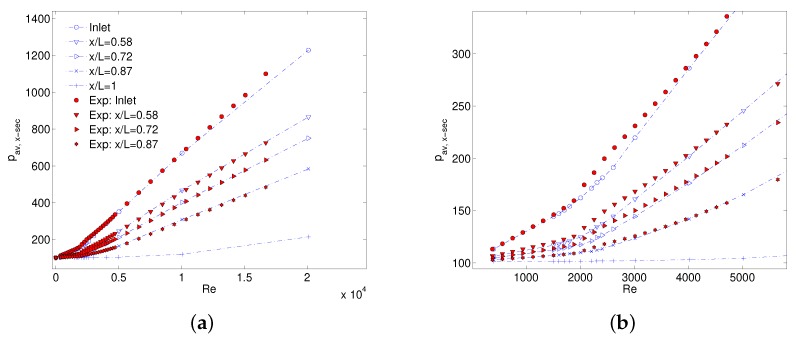
Comparison of measured and numerical static pressure inside MC1 (**a**), and zoomed low Re region (**b**).

**Figure 11 micromachines-10-00171-f011:**
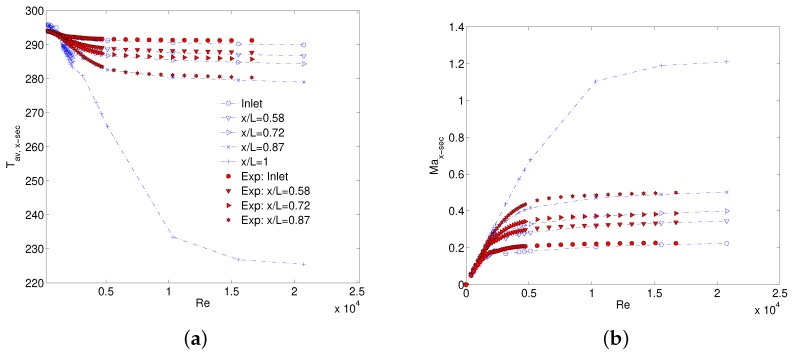
Comparison of measured and numerical static T (**a**), and Ma (**b**) inside MC1.

**Figure 12 micromachines-10-00171-f012:**
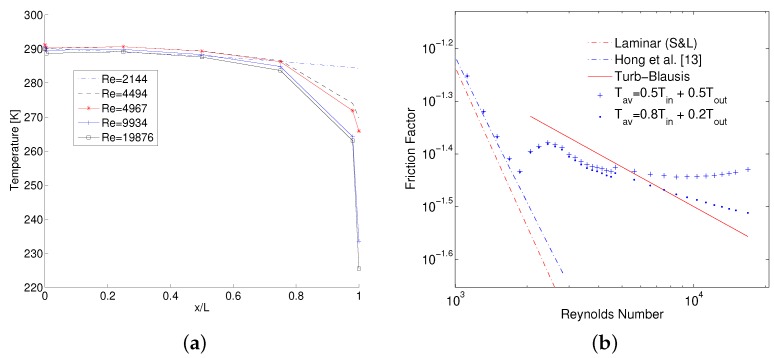
Numerical Temperature decrease along the length of MC1 at various Re (**a**), and Effect of $T_{av}$ on calculation of experimental $f_{f}$ (**b**).

**Figure 13 micromachines-10-00171-f013:**
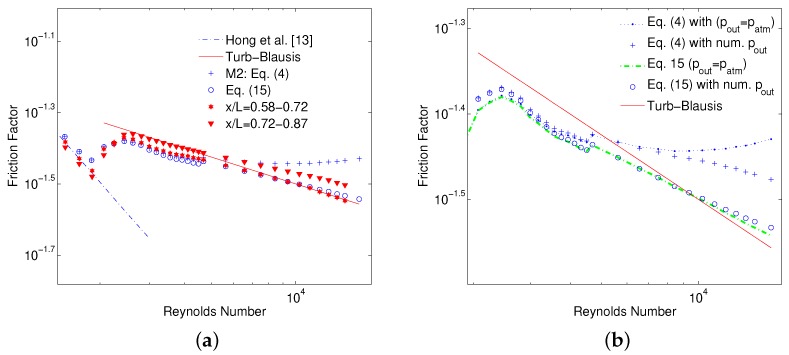
Comparison of $f_{f}$ calculation by using: M2, Equation (Equation 15) and semi-local values (**a**), Equations (Equation 4) and (Equation 15) with different outlet pressure treatment (**b**).

**Table 1 micromachines-10-00171-t001:** Typical uncertainties of instruments used.

Instrument	Range (0–Full Scale (FS))	Uncertainty
Volume flow rate controllers	0–500 & 0–5000 nmL/min	0.5% FS
Pressure sensors	0–256, 0–860 & 0–1460 kPa	0.5% FS
K-type thermocouple	0–100 ${}^{\circ }$C	0.25% FS

**Table 2 micromachines-10-00171-t002:** Data Reduction Methods Used in Current Study.

Location	M1	M2
**Inlet**	- Numerical minor loss coefficient to	- Isentropic expansion between
	estimate entrance manifold pressure drop	entrance manifold (1) and MC inlet (in)
	- $T_{in}=T_{1}$ (see Figure 3)	
**Outlet**	- Fully expanded flow ($p_{out}=p_{atm}$)	
	- $T_{out}$ estimated using Equation (Equation 5)	

**Table 3 micromachines-10-00171-t003:** Channel geometry used for simulations.

Channel	*w* (μm)	*h* (μm)	$D_{h}$ (μm)	α
MC1	360	250	∼295	0.7
MC2	1020	112.7	∼203	0.11
MC3	550	110	∼184	0.2

**Table 4 micromachines-10-00171-t004:** Boundary Conditions.

Boundary	Value
Inlet	- mass flow rate: experimental or from Equation (Equation 3)
	- Turbulence Intensity, TI = 5%
	- Temperature $T_{in}=23$ ${}^{\circ }$C
Walls	- No slip
	- Adiabatic
Outlet	Pressure outlet, Relative p = 0 Pa

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
