# Peer review of "A Comparison of Data Reduction Methods for Average Friction Factor Calculation of Adiabatic Gas Flows in Microchannels"

_micromachines, 2019, doi:10.3390/mi10030171_

Round 1

Reviewer 1 Report

In the original response to the article, several major and minor points were raised. In their response, the authors have addressed all concerns and updated their manuscript. Therefore, I’d like to recommend this article for publication.

Reviewer 2 Report

In the revised manuscript the authors have made sufficient changes to satisfy the concerns of this reviewer. The article is recommended for publication.

This manuscript is a resubmission of an earlier submission. The following is a list of the peer review reports and author responses from that submission.

Round 1

Reviewer 1 Report

In this manuscript the authors provide a guideline for experimentally estimating the friction factor for flow of gasses in microchannels. The methodology employs both experimental and numerical approach. While the work is detailed following suggestions are made to improve the understading of the readers.

The title of the manuscript can be improved, the present title lays emphasis on the discussion portion of the paper.

Sensitivity and range of instrumentation used in experimental work should be presented in the form of a table.

Sufficient geometric detail should be provided in figure 4 for the readers to clearly understand the computational domain.

Detail mesh parameters (aspect ratio, orthogonality, skewness) should be provided (preferably in the form of a table) for each microchannel geometry.

Author Response

Answers to the reviewers' comments are attached in the word file.

Reviewer 2 Report

Rehman et al. described three microchannels for the model-experiment comparison such that agreement between them could be assessed gas flow characteristics. This work seems to be an extension of previously published work by Hong et al. (2016). The new contribution seems to be a PMMA plastic sheet instead of a silicon wafer and minor loss coefficients in addition to friction factors. Overall, the work is not put in context very clearly and efficiently. Frankly, understanding this manuscript requires readers to read some references in advance, especially Hong et al., 2016. All in my question would be what is the significant contribution that this article brings that wasn’t already present in Hong et al., 2016.

It is not clearly explained why authors used three different MCs. Did authors try to obtain different Dh and AR?  If so, why are all different values for w and h selected? It seems that only three hydraulic diameters and aspect ratios are tested, not in the range of 184 – 295 micrometer and 0.11 – 0.7, respectively. Does different minor loss coefficients shown in Fig. 6 come from different hydraulic diameter or aspect ratio?

In line 372, what is isothermal assumption? In Fig. 8, which part or line represents the isothermal case? Similarly, in line 375, isothermal friction factor is not clearly defined. It seems that eq. 14 is for adiabatic MCs. In line 378, does available data reduction methodologies mean M1 and M2?

The average roughness of the MC is 1.05 micrometer and its relative roughness is less than 0.5 %. Does other roughness larger than 1 or 2 % affect experimental friction factor?

Authors barely talked about M3? Why is it included?

Considering all these, the importance of the work is greatly weakened. Besides, there are some other major revisions needed:

In line 72 and 79, there is an additional or missing round bracket.

In line 74-75, is N ml/min a unit for mass flow rate?

Below line 114 (in eq. 4), there is no explanation about L.

In line 129, minor loss is (ΔPin/ ΔPout,) while, below line 134, minor loss coefficient is ΔPin/out. Which one is correct?

Below line 134 (in eq. 7), there is no clear explanation about Kin/out, ΔPin/out, and uin/out, even in the abbreviations section.

Below line 141 (in eq. 8), what does T1 mean?

Below line 178 (in table 2) and line 188, please use a consistent notation for an aspect ratio (AR or alpha).

Blow line 192 (in Fig. 5), there are two same lines (red dash-dot lines) from laminar (Shah&London) and Turb-Blausis? Are they perfectly identical?

Above line 193, the location of eq. 12 seems to be changed. What does fRe mean? It seems to be the product of friction factor and Reynolds number which is missing in the manuscript.

Below line 219 (in eq. 13), what does subscript “th” mean?

Below line 243 (in Fig. 7), where is an equation or other source for “isothermal” line?

In line 252, it seems that there is no explanation about epsilon in the manuscript.

Above line 257 (in Fig. 8b), a legend is missing. It seems to be a open circle instead of a open triangle.

In line 270, there is a lower case kout which is not defined in the manuscript.

Below line 317 (in Fig. 9), based on your abbreviations, it seems that Maout must be used.

Below line 329 (in Fig. 10), based on your abbreviations, it seems that P must be replaced with p.

Below lien 337 (in Fig. 11), it seems that M must be replaced with Ma.

IN line 359, there is another alpha in addition to aspect ratio.

Several typos:

In line 4, “upto” seems to be “up to.”

In line 82, mass flow rate seems to be “Mass flow rate”.

Above line 360 (in Fig. 13), there are two “M2’s.” Does M2 have two equations?

Author Response

(The authors gave the same response as above.)

Reviewer 3 Report

This is a high quality paper. The paper title clearly describes content of the work. The paper is well written and the quality of the figures is acceptable. The chosen references are adequate for the topic of the paper and cover recent important papers. The methods are correct and very well written and detailed. The paper is interesting with several important conclusions and recommended for publication in this journal.

Author Response

Answers to reviewers' comments are attached in the word file.
